# The Combination of Neurotropic Vitamins B1, B6, and B12 Enhances Neural Cell Maturation and Connectivity Superior to Single B Vitamins

**DOI:** 10.3390/cells14070477

**Published:** 2025-03-22

**Authors:** Oscar Cuyubamba, Camila Pereira Braga, Dionne Swift, John T. Stickney, Christian Viel

**Affiliations:** 1The Procter & Gamble Company, Mason Business and Innovation Center, 8700 Mason Montgomery Road, Mason, OH 45040, USA; cuyubamba.oa@pg.com (O.C.); pereirabraga.c.1@pg.com (C.P.B.);; 2P&G Health Germany GmbH, German Innovation Center, Sulzbacher Straße 40, 65824 Schwalbach am Taunus, Germany

**Keywords:** neurotropic B vitamins, nerve regeneration, peripheral neuropathy, proteomics

## Abstract

Peripheral neuropathy (PN) is a prevalent condition characterized by damage to peripheral nerves, often linked to risk factors such as diabetes. This condition results from various forms of neural damage, including injury to the cell body, axons, or demyelination, frequently beginning with small and thinly or unmyelinated fibers. Such nerve damage disrupts normal signaling, leading to symptoms like numbness, tingling, and pain. Effective nerve repair and regeneration, particularly through remyelination, are essential therapeutic objectives. While vitamin B12’s role in repair processes has been well established, emerging evidence suggests that other neurotropic vitamins, specifically B1 and B6, also contribute significantly to nerve health and symptom relief in PN. In this study, we demonstrate that a combination treatment of vitamins B1, B6, and B12 enhances repair and oxidative stress responses in co-cultures of neural and Schwann cells, leading to improved cell maturation and connectivity compared to vitamin B12 alone. Furthermore, proteomic analysis supports these observations at the molecular level, with enhanced cellular recycling processes like proteasome enhancement, as well as protein synthesis upregulation, needed to rebuild nerve connections and combatting oxidative stress. Our combined morphological and molecular results highlight the potential therapeutic advantage of the B1, B6, and B12 combination over vitamin B12 alone.

## 1. Introduction

Peripheral neuropathy (PN), a condition characterized by damage to the peripheral nerves, is a major clinical concern due to its prevalence and significant impact on patients’ quality of life [1,2,3]. Since diabetes mellitus is the most identifiable cause of PN, DPN (diabetic peripheral neuropathy) is the most common form of PN, accounting for up to 50% of cases [1,4,5]. Notably, idiopathic peripheral neuropathy, where no clear cause can be identified, accounts for up to 25% of PN cases. Besides diabetes mellitus, causes of PN are manifold and include vitamin deficiencies (e.g., vitamin B1, B6, B12), certain medications, obesity, hereditary disorders, inflammatory and infectious diseases, neoplasms, toxic agents, and alcohol abuse [1,6,7]. PN is caused by neuronal damage on the cell body or axons, demyelination, or a combination of those [8]. Often small fiber damage occurs first, including the peripheral thinly myelinated Aδ fibers as well as unmyelinated C nerve fibers, leading to small fiber neuropathy [9]. PN causes various symptoms, such as tingling, numbness, burning, pins and needles sensation and stabbing pain, generally in the hands and feet, getting worse at night. Symptoms of PN are often difficult to manage and have a significant impact on all aspects of patients’ lives. The etiology of PN is multifactorial, involving metabolic, toxic, inflammatory, and genetic factors [1,8]. Among the numerous strategies proposed to prevent or alleviate PN, the role of neurotropic B vitamins—specifically vitamins B1 (thiamine), B6 (pyridoxine), and B12 (cobalamin)—has received considerable attention due to their essential roles in maintaining nerve health and function [6,9,10,11] and has been an established treatment strategy in clinical practice for decades.

### 1.1. Vitamin B1 (Thiamine) and Nerve Health

Vitamin B1, or thiamine, is a critical cofactor in carbohydrate metabolism, playing a pivotal role in the generation of adenosine triphosphate (ATP) through the pentose phosphate pathway and the Krebs cycle [10,12]. This energy production is crucial for nerve function, as neurons have high energy demands. Thiamine deficiency can lead to significantly impaired nerve function and neurological deficits, including Wernicke’s encephalopathy and beriberi, conditions marked by nerve degeneration. In the context of peripheral nerves, thiamine deficiency impairs axonal conduction and myelin maintenance, leading to neuropathic symptoms [10,12].

Research has demonstrated that thiamine supplementation can ameliorate symptoms of diabetic neuropathy by improving nerve conduction velocity and reducing oxidative stress, which is often elevated in neuropathic conditions. The antioxidant properties of thiamine are particularly important in protecting nerves from the damaging effects of hyperglycemia-induced oxidative stress, a common pathway in diabetic neuropathy. Thus, maintaining adequate thiamine levels is essential for preventing nerve damage and supporting nerve regeneration [10,12].

### 1.2. Vitamin B6 (Pyridoxine) and Neurotransmitter Synthesis

Vitamin B6, or pyridoxine, is involved in the biosynthesis of neurotransmitters such as serotonin, dopamine, and gamma-aminobutyric acid (GABA), all of which are vital for proper nerve function. Pyridoxine also plays a role in amino acid metabolism, hemoglobin production, and gene expression [10,12].

Pyridoxine deficiency is associated with symptoms such as irritability, depression, and cognitive decline, which are often due to impaired neurotransmitter synthesis. In peripheral nerves, insufficient vitamin B6 can result in defective myelin formation, leading to demyelination and subsequent neuropathy. Clinical studies have shown that vitamin B6 supplementation can reduce neuropathic pain, particularly in conditions such as carpal tunnel syndrome and diabetic neuropathy [10,12].

### 1.3. Vitamin B12 (Cobalamin) and Myelin Integrity

Vitamin B12, or cobalamin, is essential for the maintenance of myelin, the protective sheath that surrounds nerve fibers. It also plays a crucial role in DNA synthesis and methylation processes, which are important for cellular repair and regeneration [10,12]. A deficiency in vitamin B12 can lead to a range of neurological disorders, including subacute combined degeneration of the spinal cord [10], which is characterized by demyelination of the dorsal columns and corticospinal tracts.

PN is a common manifestation of vitamin B12 deficiency, particularly in elderly populations and individuals with malabsorption syndromes, such as pernicious anemia [10]. The pathophysiology of B12 deficiency-induced neuropathy involves impaired myelin formation, axonal degeneration, and neuronal death. Clinical studies have demonstrated that vitamin B12 supplementation can significantly improve symptoms of neuropathy, such as paresthesia and motor deficits, by promoting remyelination and enhancing nerve regeneration [10,12].

Moreover, the neuroprotective effects of vitamin B12 extend beyond its role in myelination. Cobalamin has been shown to enhance the synthesis of nerve growth factor (NGF), a critical protein involved in the survival and maintenance of sensory and sympathetic neurons. This neurotrophic effect underscores the importance of vitamin B12 in not only preventing nerve damage but also facilitating the repair of injured nerves [10].

### 1.4. Synergistic Effects of B Vitamins

While each of these B vitamins plays a distinct role in nerve health via a unique mode of action and cannot replace each other in specific biochemical pathways, their synergistic effects are particularly noteworthy [10,13]. Vitamins B1, B6, and B12 often work together in biochemical pathways that support nerve function. For instance, thiamine and cobalamin are both involved in energy metabolism, while pyridoxine and cobalamin are crucial for neurotransmitter synthesis and myelin integrity [10]. The combined supplementation of these vitamins has been shown to provide enhanced neuroprotective effects compared to individual vitamin supplementation [12], suggesting that a comprehensive approach to B vitamin therapy could be more effective in preventing and treating PNs of different etiologies, such as DPN and idiopathic peripheral neuropathy.

The aim of this study is to explore the synergistic effects of the combination of vitamins B1, B6, and B12 on nerve maturation and connectivity, comparing these outcomes to the effects of each vitamin individually, with a focus on determining the benefits attributed to the combination of all three neurotropic B vitamins in promoting neural health and functionality. Additionally, using in vitro methodology to simulate and investigate the function of the nervous system is an innovative approach to avoid animal experiments.

Our research demonstrates that combination treatment with B1, B6, and B12 yields better results in protection against oxidative stress, driving neural maturation and cellular connectivity vs. mono vitamin treatment alone.

## 2. Materials and Methods

### 2.1. Cell Cultures

SCL4.1 cells (immortalized rat Schwann cell line) [14] and NG108 cells (mouse neuroblastoma × rat glioma hybrid neural cell line) were cultured per directions from ATCC (Manassas, VA, USA, NG108-15 [108CC15] HB-12317™ product sheet) in their respective media. Specific cell information is available on the supplier websites (93031204, rat nerve (sheath), Flattened or crescent shaped | Sigma-Aldrich). Cells were plated at a 2.0 × 10^6^ density on a poly-D-lysine coated (75 µg mL^−1^, Sigma–Aldrich) 96-well, black-walled microplate with clear bottom (#655 090; Greiner Bio-One, Gremsmuenster, Austria) in BrainPhys Neuronal serum free medium (Stemcell Technologies, Vancouver, BC, Canada) (100 µL per well) containing 10% FBS and 1% penicillin/streptomycin 24 h prior to assaying.

#### 2.1.1. 3D Co-Culture Model

Matrigel^®^ basement membrane matrix (Corning, Corning, NY, USA) was thawed overnight in a 4 °C refrigerator. On day 0, the Matrigel matrix was diluted to 5 mg/mL with ice-cold SCL4.1/NG108 complete cell culture medium, and 5 µL was added into each well of a pre-chilled 96-well plate, spread evenly with a pipet tip, and incubated at 37 °C for 15–20 min to form a gel. We then trypsinized the SCL4.1 and NG108 cells, pelleted them, and resuspended them to a density of 5 × 10^6^ cells/mL. Next, 10 µL of each cell line was mixed with 70 µL matrix and 60 µL medium and given to each well of the loaded 96-well plate (Figure 1A). The plates were incubated at 37 °C for 30–45 min to establish the culture. The culture was kept at 37 °C. Every two days, the medium was changed. The treatments were incorporated with media changes 1 day after 3D co-culture establishment.

Colorimetric and fluorometric analyses of 3D and 2D co-cultures and observing cell morphology were done using light and phase contrast microscopy, respectively.

#### 2.1.2. 2D Co-Culture Model for Synapsing and Networking Measurements

For 2D neural/Schwann cell co-culture model creation, to assess synapsing and networking parameters, we trypsinized NG108 and SLC4.1 cells and co-plated each at a density of 10,000 cells/well in a 96-well plate (Figure 1B). The co-cultures were incubated in a CO_2_-incubator for 24 h before application of treatments/controls/stressors. We chose 2D co-cultures to facilitate connectivity measurement.

### 2.2. Preparation of Test Medium

Gibco (Thermo Fisher, Waltham, MA, USA) DMEM: F12 medium with low-level vitamin B1, vitamin B6, and vitamin B12 (6.4 µM thiamine hydrochloride, 9.8 µM pyridoxine hydrochloride and 0.05 µM B12; referred to as ‘control’) was prepared using a method similar to what was described previously. This low-level control medium represents a state of non-deficient B vitamin levels and mimics physiological conditions. In contrast, vitamins B1, B6, and B12 were added individually or in combination (referred to as ‘treatment’) to prepare the test media. Concentrations were 40 μM for vitamin B1, 20 μM for vitamin B6, and/or 0.4 μM for vitamin B12 [15,16].

### 2.3. Insult to Induce Nerve Cell Damage

To induce cellular damage (referred to as ‘insult’), we used 200 µM hydrogen peroxide (H_2_O_2_) for 6 h, as oxidative stress is one of the dominant mechanisms in the pathophysiology of PN. Following the insult, the medium was removed and replaced with either control medium or treatment medium.

### 2.4. Analytical Assays

#### 2.4.1. AlamarBlue^®^ Cell Viability Assay

To measure metabolic reduction and quantify cell proliferation and health in the 3D co-culture, we used the alamarBlue^®^ assay (Invitrogen, Waltham, MA, USA) as described by the manufacturer. Co-cultures of neurons and Schwann cells in 3D matrix were assessed with alamarBlue^®^ for viability, both in non-insulted culture and after insult via the application of hydrogen peroxide. We placed 2.5 × 10^5^ NG108 and 2.5 × 10^5^ SCL4.1 cells in each well of a 96-well plate. After 24 h, treatment and insult were applied with media change. The treatment plate was incubated for 6 h, then the final 10 µL alamarBlue^®^ cell viability reagent (DAL1025 10 × solution) was added to each well of the 96-well plate and incubated for 2 h. Fluorescence was read using a Synergy Neo plate reader (Agilent, Santa Clara, CA, USA) set to 560/590 nm (excitation/emission).

#### 2.4.2. NeuroFluor™ Maturation Detection Assay

The NeuroFluor™ NeuO dye (Stemcell Technologies) was employed to assess whether vitamin treatment had any impact on neuron maturation in the 3D neuron/Schwann cell cultures. NeuroFluor is a membrane-permeable fluorescent probe that selectively labels primary and pluripotent stem cell-derived neurons in live cultures and fluoresces green when bound to a mature neuron. Cells labeled with NeuroFluor can be visualized using fluorescent imaging. As with the alamarBlue assay, 3D cultures were treated with the insult (H_2_O_2_) and then with individual B vitamins or the combination of all three neurotropic B vitamins, then incubated for 6 h before adding 1 µL NeuroFluor to each well of the 96-well plate. Fluorescence was read using a Synergy Neo plate reader set to 468/557 nm (excitation/emission). Images of cultures stained with NeuroFluor were obtained using an inverted Keyence BZ-X810 microscope (Keyence, Osaka, Japan).

#### 2.4.3. Neuron 2D Cell Culture Morphology

Cultured NG108 neural cells were monitored using the Keyence BZ-X810 microscope for live cell analysis, and images were captured every 6, 12 and/or 24 h. Final recordings were measured and analyzed with regard to morphological features, neural distances, cell numbers, clumps, associated neuronal sets (networking), and networking sets. Captured images were framed in 1.5 mm × 1.5 mm areas. In every frame, cells and all dendrites per cell were counted and measured. Dendrite measurement was performed by tracing lines along the dendrite extensions using the microscope images scale.

#### 2.4.4. 2D Neuron Images Masking Methodology

The image contrast was increased by 30% and sharpened by 45% to improve visualization. The neuron cell bodies were rendered and isolated, then registered as orange circles using the Keyence BZ-X810 microscope. Dendrites were masked as blue lines with an assigned threshold thickness of 0.06 mm. Once the masking was optimized to fit the cell bodies and dendrites, an algorithm was created to count synapses as dendrites extended from cell to cell. Networking was defined as the establishment of groups of three or more cells, including synaptic connections, which were defined as nodes. The number of nodes was then counted.

### 2.5. Statistical Analysis

Normality tests were conducted on all data to determine appropriate statistical tests, and a one-way analysis of variance (ANOVA) followed by a post-hoc test was performed. * *p* < 0.05, ** *p* < 0.01, *** *p* < 0.001 were considered to be significant. Data are expressed as means ± SEM when described in the text.

### 2.6. Proteomic Analysis

#### 2.6.1. 3D Co-Culture Sample Preparation and Data Acquisition for Proteomic Analysis

To maintain protein stability, 96-well plates were placed on dry ice. Next, a lysis buffer containing 7M urea, 30 mM Tris, and 4% CHAPS was added to each well. A volume corresponding to 50 μg of protein was transferred to a new tube and mixed with 4 volumes of cold acetone, followed by incubation on ice for 1 h. After incubation, samples were pelleted for 10 min at 15,000× *g* and the supernatant was discarded. The excess acetone was evaporated at room temperature for approximately 15 min. The dried delipidated pellets, containing 50 μg of protein, were resuspended in 20 μL of denaturation buffer (25 mM ammonium bicarbonate (ABC) at pH 8.0, 10 mM Tris (2-carboxyethyl) phosphine) (TCEP), 5% sodium deoxycholate (SDC)). To fully solubilize the protein, the samples were incubated at 60 °C for 10 min. Next, 5 μL of alkylation buffer (100 mM iodoacetamide in water) was added to the samples. The samples were then incubated in the dark for 60 min at room temperature. To dilute the samples and prepare them for further processing, 175 μL of dilution buffer (25 mM ABC, pH 8.0) was added. Additionally, 2 μL of trypsin solution (1 μg/mL in 25 mM ABC) was added to the samples. The samples were then incubated in the dark at 37 °C overnight to allow for enzymatic digestion. To quench the reaction and remove SDC, 10 μL of 10% trifluoroacetic acid (TFA) was added. This step also lowered the pH to 5.0, causing SDC to precipitate. After 30 min incubation, the pelleted SDC was removed by centrifugation at 15,000× *g* for 10 min. The resulting supernatants were carefully transferred to new tubes for direct analysis by Liquid Chromatography-Trapped Ion Mobility Spectrometry-Mass Spectrometry (LC-TIMS-MS/MS). LC-TIMS-MS/MS was carried out using a nanoElute UHPLC System (Bruker Daltonics, Bremen, Germany) coupled to the timsTOF Flex mass spectrometer (Bruker Daltonics) equipped with a CaptiveSpray ion source (Bruker Daltonics). Samples were loaded on a C18 analytical PepSep nanocolumn (25 cm length, 150 μm inner diameter, 1.5 μm particle size; Bruker Daltonics) and eluted using a linear gradient between 0.1% formic acid in water and 0.1% formic acid in acetonitrile. The column was kept at 55 °C using Column Toaster (Bruker Daltonics) during the entire analysis.

#### 2.6.2. Proteomics Data Processing and Theme Analysis

MS raw files were processed with MaxQuant (Max Planck Institute of Biochemistry, Martinsried, Germany). MS/MS spectra were matched to in-silico-derived tryptic peptide fragment mass values from the UniProt Human database [17]. Label-free quantification (LFQ) was performed using classic normalization. A subsequent thematic analysis was performed on the proteins that showed significant enrichment following the triple vitamin combo treatment compared to those treated solely with vitamin B12. After MaxQuant processing, data were filtered to remove potential contaminants, and proteins that were present in at least 50% of the samples were considered valid. After cleaning the data and imputing missing values, the intensities of the proteins were transformed using a logarithm base 2 and used for statistical analysis. Volcano plots of differentially regulated proteins and various graphs were generated by R [18]. A protein was considered differentially expressed when exhibiting a >1.5-fold change and a *p*-value < 0.1. To conduct functional enrichment analysis, we utilized the ShinyGO 0.77 (South Dakota State University, Brookings, SD; USA) and Enrichr (Icahn School of Medicine at Mount Sinai, New York, NY, USA) tools to identify the biological processes in which the proteins were involved.

## 3. Results

A 3D and a 2D neural cell co-culture model has been used to investigate different research questions. The 3D co-culture model of neurodegeneration was established before by Smith and co-workers [16]. The effects of the single B vitamins were not studied prior to their work, nor was the combination of neurotropic B vitamins compared with a treatment of single B vitamins. Instead, the effects of the combination of neurotropic B vitamins were compared to no treatment controls. In contrast to this work, Smith and co-workers used a custom-made medium to culture the cells that did not contain any vitamin B1, B6, or B12, representing a depleted state [16]. In the current study, a very low concentration of B vitamins was considered appropriate as the control medium because complete depletion of B vitamins is not believed to occur in a physiological environment.

### 3.1. The Combination of the Vitamins B1, B6, and B12 Is Superior in Supporting Nerve Cell Viability in Healthy Cells and Regeneration After Damage (3D Co-Culture Model)

We used hydrogen peroxide to stimulate oxidative damage as a model of one of the underlying mechanisms of PN. Co-cultures of neurons and Schwann cells in a 3D matrix were assessed for viability both in a non-insulted culture, to investigate the effect of B vitamins on healthy cells and under oxidative stress conditions (hydrogen peroxide-insulted) and to investigate the restorative effect of B vitamins on the cells after nerve damage. To visualize the cells, alamarBlue/resazurin cell viability assay was used. The cytotoxicity of the hydrogen peroxide insult is demonstrated when comparing the controls in Figure 2 (red bars). Fluorescence units are lower in Figure 2b, showing the harmful effect of hydrogen peroxide on cell viability. As Figure 2a shows, vitamin B1 (+62.29%; *p* < 0.01) and B6 (+55.15%; *p* < 0.01) significantly enhanced cell viability in healthy cells when compared to the control, while the effect was insignificant for vitamin B12 alone (+3.70%; *p* > 0.05). Treatment with the combination of vitamins B1, B6, and B12 provided a highly significant increase (+97.32%; *p* < 0.001) compared to the untreated control (Figure 2). The combination of vitamins B1, B6, and B12 was 26.34-fold more effective in increasing cell viability than the B12 treatment alone. The increase was statistically significant (*p* < 0.01). Vitamins B1 and B6 also elevated cell viability (1.56-fold for B1 treatment and 1.76-fold for B6 treatment, respectively), but the effects were not statistically significant. Results were similar for the damaged cells after the insult as shown in Figure 2b. All B vitamins lead to a significant increase in cell viability compared to the untreated control. However, cell viability increased the most (+59.23%; *p* < 0.01) with the combination of vitamins B1, B6, and B12 when compared to the untreated control. Vitamin B1 significantly elevated cell viability by 40.31% (*p* < 0.01), vitamin B6 by 32.17% (*p* < 0.05), and vitamin B12 by 41.45% (*p* < 0.01). The combination of vitamins B1, B6, and B12 elevated cell viability 12.57% more effectively than the treatment with vitamin B12 alone. This increase was statistically significant (*p* < 0.05). Compared to the treatment with vitamins B1 and B6, the treatment with the combination of all three neurotropic B vitamins was 13.48% and 20.47% more effective, respectively.

This observation is in line with the study objective, i.e., to confirm the biochemical synergy for the combination treatment reported in the literature. This effect arises as a result of converging biochemical pathways which interconnect the neurotropic B vitamins B1, B6, and B12.

### 3.2. The Combination of the Vitamins B1, B6, and B12 Is Superior in Supporting Cell Maturation (3D Co-Culture Model)

When assaying for the ability to induce cell maturation, we observed highly significant synergistic effects for the combination of vitamins B1, B6, and B12 compared to the untreated control (Figure 3). Cell maturation was 50.49-fold increased (*p* < 0.001) with the combination of vitamins B1, B6, and B12 in comparison to the untreated control (Table 1). Treatment with individual B vitamins increased cell maturation 4.46-fold for vitamin B1 (*p* < 0.001), 1.97-fold (*p* < 0.001) for vitamin B6, and 18.17-fold (*p* < 0.001) for vitamin B12. Comparing the effect of the combination of vitamins B1, B6, and B12 on cell maturation to the individual B vitamin treatments, the combination was 2.88-fold more effective in supporting cell maturation than B12 treatment, 50.90-fold more effective than B6 treatment, and 14.31-fold more effective than B1 treatment. All of these effects were highly significant. The effects of combination treatment were highly significant and confirm a synergistic effect on nerve cell maturation that only occurs when all three neurotropic B vitamins are present.

### 3.3. The Combination of the Vitamins B1, B6, and B12 Is Superior in Promoting Cell Connectivity (2D Culture Model)

To further explore the synergistic effect in regard to cell function and connectivity of the combination of B1, B6, and B12, the 2D co-cultures of neural and Schwann cells along with image processing were used, both with healthy cells without insult and insulted cells after hydrogen peroxide administration. Synaptic connectivity and networking (the established group of three or more neurons connected via synaptic connections presenting at least a node) were measured. Synapses and networks were counted as frequency (events per total cells). Micrographs were taken in phase contrast on a 10-fold magnification of both the control neurons and vitamin B combination treated neurons. Rendering and image processing was added (coloring, define lines and cell bodies), as well as cell masking, for better analysis. Approximately 220 cells per field of view per well were measured. The cytotoxicity of the hydrogen peroxide insult is demonstrated when comparing the controls in Figure 4 (red bars). Frequency occurrence is lower in Figure 4b, showing the harmful effect of hydrogen peroxide on cell connectivity. The results shown in Figure 4a demonstrate a significant effect of the combination treatment of vitamins B1, B6, and B12 in regard to synapsing on healthy cells when compared to untreated cells (4-fold increase; *p* < 0.01). The vitamin B12 treatment increased synapsing of healthy cells 2.55-fold in comparison to untreated cells (*p* < 0.01). In direct comparison, the combination of vitamins B1, B6, and B12 supported synapsing 1.94-fold more effectively than treatment with vitamin B12 alone (*p* < 0.05). With insulted cells (Figure 4b), the same pattern was observed whereas levels of significance were even higher, reconfirming the superior effect of the combination of vitamin B1, B6, and B12 to individual B vitamin treatment in regard to regeneration. The increase in synapsing was highly significant, and was 5.44-fold (*p* < 0.001) higher for cells treated with the combination of vitamins B1, B6, and B12 than in untreated cells. The vitamin B12 treatment increased synapsing 3-fold compared to untreated cells. This increase was statistically significant (*p* < 0.01). In comparison to vitamin B12 treatment, the combination of vitamins B1, B6, and B12 was 2.22-fold more effective in supporting synapsing after the insult with hydrogen peroxide (*p* < 0.01) Table 2 provides an overview of the extent of the treatment effects vs. control and vs. individual B vitamin treatments in regard to cell connectivity.

Both vitamin B12 alone and the combination of vitamins B1, B6, and B12 significantly enhanced synapsing and the interaction of neural cells and Schwann cell connectivity compared to the untreated control. However, the combination of vitamins B1, B6, and B12 demonstrated a superior effect to treatment with only vitamin B12 in regard to synapsing, which was statistically significant regardless of whether cells were insulted or not. The effects of vitamin B1 and vitamin B6 have not been investigated in this experimental setup (Figure 4).

The results shown in Figure 5a from healthy cells demonstrate a significant effect of the combination of vitamins B1, B6, and B12 in regard to networking when compared to untreated cells (4.6-fold increase; *p* < 0.001). The vitamin B12 treatment increased networking 3.2-fold compared to untreated cells (*p* < 0.01). The combination of vitamins B1, B6, and B12 was 1.64-fold more efficient in promoting networking than the treatment with only vitamin B12 (*p* < 0.05 with insulted cells (Figure 5b)). This is further illustrated in Figure 5c, which shows non-insulted untreated cells with fewer connections.

The same pattern was observed, in which levels of significance were even higher, reconfirming the superior effect of the combination of vitamins B1, B6, and B12 to individual B vitamin treatment in regard to regeneration. The increase in networking was 6.44-fold (*p* < 0.001) higher for cells treated with the combination of vitamins B1, B6, and B12 than in untreated cells. Treatment with vitamin B12 increased networking 3.44-fold in comparison to untreated cells (*p* < 0.01). The combination of vitamins B1, B6, and B12 was 2.23-fold more efficient in promoting networking than the treatment with only vitamin B12 (*p* < 0.01). Figure 5d reflects how insulted cells treated with the combination show more connections compared to those in Figure 5c.

### 3.4. Proteomics Analysis Provides Molecular Support for the Enhancement of Connectivity and Resistance to Oxidative Stress by the Combination of the Vitamins B1, B6, B12

To better understand the cellular mechanisms that were involved in promoting nerve cell regeneration after insult, neuronal maturation, and cellular connectivity, a proteomics assessment was conducted.

Table 3 describes the top themes when comparing the combination of vitamin B1, B6, and B12 treatment to treatment with vitamin B12 alone. We observed an enhancement of proteasome, ribosome, and oxidative stress recovery pathways when treating the 3D co-culture with the combination of vitamins B1, B6, and B12. Many of these top themes underscore the findings that the combination of vitamin B1, B6, and B12 enhances neuronal survival, regeneration, maturation, synapsing, and networking to a higher extent than treatment with vitamin B12 alone, as Figure 2, Figure 3, Figure 4 and Figure 5 show.

In the principal component analysis (PCA) plot (Figure 6a), the distinct clustering of data points reflects a clear separation among the groups, indicating significant differences in their overall protein expression profiles. Each group forms a separate cluster, suggesting that the treatments or conditions being compared have led to observable and distinguishable changes in protein levels. Following this, the volcano plots (Figure 6b) reveal the proteins that exhibit statistically significant differences across various comparisons. Notably, in the comparison between the combination treatment and B12 alone, proteins were found to be upregulated related to cellular signaling, protein synthesis, and oxidative stress, highlighting the potential biochemical mechanisms through which the combination treatment exerts its effects compared to B12 alone. This analysis not only underscores the efficacy of the combination treatment but also provides insights into the underlying biochemical mode of action that may drive its therapeutic benefits.

In Table A1, we present a comprehensive overview of the biochemical processes associated with specific proteins that exhibit statistically significant differences in expression levels. The table highlights both upregulated and downregulated proteins. Each entry includes the protein name, its corresponding biochemical process, and the statistical significance of the expression change, measured through *p*-values and fold changes. This structured presentation allows for a clear comparison of how these proteins are involved in critical cellular functions, such as mRNA processing, protein synthesis, redox balance, and cellular signaling, emphasizing their roles in neuronal and Schwann cell adaptation and function.

## 4. Discussion

The main objective of this study was to investigate treatment effects of individual B vitamins in comparison to a combination treatment of vitamins B1, B6, and B12 and describe outcomes of various parameters. Both biochemical and morphological parameters were investigated, and proteomic analysis was performed to obtain detailed insights into the underlying mechanisms.

Although the pathophysiological mechanisms of DPN are not entirely understood, certain pathways have been identified in preclinical studies driving harmful metabolic processes. These pathways are associated with hyperglycemia, dyslipidemia, and microvascular disease, and cause downstream metabolic disturbances, such as oxidative stress, eventually leading to neuronal cell death [19]. While the risk factor with the strongest association for developing DPN is poor glycemic control [20,21,22], other independent risk factors like obesity, an increased waist-to-hip ratio, hypertension, dyslipidemia, hypertriglyceridemia and smoking are well-described [20,21,23,24,25,26]. One of the most prominent and extensively studied pathways in DPN pathophysiology is the polyol pathway, which generates an excess of ROS with an oversupply of substrate present during hyperglycemia [27]. When abundant fructose-6-phosphate, a byproduct of glycolysis, enters the hexosamine pathway, the increased synthesis of uridine diphosphate N-acetylglucosamine (UDP-GlcNAc) in turn enhances inflammation and oxidative stress in neurons [28]. As the upstream causes may be diverse, they converge into a common downstream metabolic phenomenon: increased oxidative stress. This allows for a simplified, yet relevant experimental design, as realized by Smith et al. [16], which focused on neurite degeneration by oxidative stress and potential regenerative effects of B vitamins. The widely recognized hypothesis which justifies the use of the in vitro model that has been described here is based on the concept that oxidative stress is a major factor which contributes to the pathophysiological mechanism of PN. Besides DPN, in which the pathways leading to increased oxidative stress are well-described, oxidative stress also plays a prominent role in PN of etiologies other than diabetes mellitus, e.g., in chemotherapy-induced PN (CIPN) and alcohol-induced PN (AIPN). Chemotherapeutic agents damage neuronal mitochondria and increase the production of reactive oxygen species (ROS). Elevated ROSs lead to oxidative stress, damaging cellular components, disrupting calcium homeostasis, and activating apoptotic pathways, ultimately resulting in neuronal degeneration [29,30]. In AIPN, multiple detrimental pathways involving oxidative stress and leading to neuronal damage from free radicals and inflammatory cytokines have been described [31,32].

Using the 3D co-culture model already established by Smith and co-workers [16], we not only replicated earlier findings, but also extended the study to also investigate the effects of the individual vitamins and compared them to the combination of vitamins B1, B6, and B12. With this study, we also build on the recent findings of Rayner et al. 2025 [33]. The combination of the three neurotropic B vitamins was superior to individual B vitamin treatment in promoting neurite outgrowth in vitro. The here presented in vitro model is a simplified way to mimic peripheral neuropathy based on the most prominent and most investigated pathophysiological mechanism which is oxidative stress. In contrast to the study by Smith and Rayner [16,33], the cell cultures used as controls were not depleted of B vitamins; instead, a medium with a low concentration of B vitamins was used to resemble a physiological environment.

The alamarBlue assay demonstrated the significant superiority of the combination of vitamins B1, B6, and B12 in supporting healthy neural cells’ viability compared to the untreated control (1.973-fold, *p* < 0.001) and to the cells treated with vitamin B12 (26.34-fold, *p* < 0.01; Figure 2a). Individual treatment with vitamin B1 or B6 also significantly increased cell viability of healthy neural cells (+62.29%, *p* < 0.01 for B1 and +55.15%, *p* < 0.01 for B6), but did not reach the levels of the combination treatment. Interestingly, vitamin B12 treatment was the least effective treatment to support healthy neural cells’ viability. The energy metabolism of healthy neural cells and the amount of reducing equivalents were most effectively promoted with the combination of all three neurotropic B vitamins. This is in line with the essential biochemical role all three B vitamins have (see introduction) in energy metabolism [10,12]. When neural cells were subjected to damage via hydrogen peroxide (Figure 2b), the combination again was most effective and superior compared to treatments with individual B vitamins (1.126-fold more effective than B12, *p* < 0.05; 1.135-fold more effective than B1, *p* < 0.05 and 1.205-fold more effective than B6, *p* < 0.05). The combination of all three neurotropic B vitamins allowed the insulted cells to regenerate within 24 h to the extent of cell viability that exceeded those of individual B vitamins. In a state of high oxidative stress leading to nerve damage, the combination of vitamins B1, B6, and B12 proves to be effective in maintaining cell viability. The regenerating effect of the combination of all three neurotropic B vitamins is mainly accounted for by their biochemical function in maintaining or restoring energy metabolism and enabling cellular rebuilding [10,12]. Although anti-oxidative properties (especially for vitamin B1 [34,35]) may contribute to the beneficial effects in regenerating cell viability, they play a minor role.

The results shown in Figure 3 demonstrate the synergistic action of the vitamins B1, B6, and B12, and their superiority vs. each single vitamin in promoting neural maturation, as highlighted by the NeuroFluor staining results. The combination of vitamins B1, B6, and B12 had a 2.88-fold higher effect on neural cell maturation than the treatment with vitamin B12 alone. The effect of the combination of all three neurotropic B vitamins exceeded by far the effects of the individual B vitamins on cell maturation (14.31-fold more than B1, *p* < 0.001; 50.90-fold more than B6, *p* < 0.001), reconfirming the biochemical synergy which only unfolds when all three vitamins come together. As a marker for cell differentiation, NeuroFluor shows how important the three neurotropic B vitamins are for the development of cell-specific functions. For highly specialized cells like neurons, differentiation is paramount to exert their proper function. Cell maturation is the precondition for nerve cells to fulfil their specific function [36]. This was significantly supported by the combination of B1, B6, and B12. As cells mature, they become able to fully operate and function, which is explicitly highlighted by the results of improved cell connectivity. Figure 4 demonstrates the effects of vitamin B12 and the combination of vitamins B1, B6, and B12 on cell-specific morphological parameters. Both the development of synapses and the growth of a dendritic network are markers of interconnection, which is essential for proper neural cell function [37]. The effect of supporting synapse formation in healthy cells (Figure 4a) was strongest with the combination of all three neurotropic B vitamins compared to only vitamin B12 treatment and the control (1.94-fold vs. B12 and 4-fold vs. control, *p* < 0.05). The superior effect on synapsing of the combination of vitamins B1, B6, and B12 was sustained in damaged cells against B12 treatment and control (2.22-fold vs. B12 and 5.44-fold vs. control, *p* < 0.01; Figure 4b). Comparing cells treated with all three B vitamins, i.e., insulted and healthy cells, occurrence frequency of synapsing was even more pronounced with the damaged cells (Figure 4b). This shows that the combination of all three neurotropic B vitamins enabled the insulted cells to regenerate and build synapses of the extent of undamaged cells. The ability to form dendritic networks was significantly improved by the combination of all three neurotropic B vitamins in both healthy and insulted cells compared to the treatment with only vitamin B12 (Figure 5; 1.64-fold for healthy and 2.23-fold for insulted cells). It is noteworthy that the network formation was much more pronounced in damaged cells. It seems that the damaged cells in particular profited from the B vitamin treatment. These results indicate that the combination of all three neurotropic B vitamins is superior under physiological conditions, but the impact of the neurotropic B vitamins is amplified under stress conditions transferable to the situation associated with PN. Oxidative stress seems to be one major factor to determine PN [38,39,40,41]. As shown in this study, the combination of vitamins B1, B6, and B12 also enhanced cell-to-cell association. This process might also enhance Schwann cells to properly interact with neural cells, to help form protective myelin sheaths around neuronal extensions, and to ensure proper signal transmission [42].

The present study shows that combination treatment with vitamins B1, B6, and B12 is superior for supporting cell viability, regeneration, neuron maturation, and cell connectivity. The increased benefit of this combination compared to the individual B vitamins has already been demonstrated in animal studies at the symptomatic and functional level [43]. This study complements these in vivo findings and provides a body of evidence which confirms the biochemical synergy on a cellular level in vitro. The results also confirm the validity of this cell culture model.

Further confirmation that the combination of vitamins B1, B6, and B12 may promote these processes derives from experimental animal studies [44,45]; however, while these studies did investigate the function, they did not explicitly examine the cell-to-cell association which becomes possible with in vitro models or provide detailed explanations in line with the in vivo findings. Proper myelination is further necessary to insulate neurons and facilitate proper synapsing interactions to promote correct proper signaling. These processes are damaged under oxidative stress and by other pathophysiological factors in a variety of diseases and conditions, resulting in faulty signaling and disrupted signal conduction, which in turn leads to PN symptoms [10,12].

The structural networking and synapsing enhancement that occurs by combining all three of these B vitamins is further supported by the results of the proteomics analysis, which provided novel findings and revealed enhanced protein expression levels of proteins related to neurite extension and cellular remodeling, as well as a molecular mode of action (Figure 6). The proteins enhanced by the combination of vitamins B1, B6, and B12 versus treatment with B12 alone support the synergistic nature of the three B vitamins in stimulating protection, repair, and adaptation processes. This synergistic effect can help damaged peripheral neurons to reconnect and restore proper nerve function and signaling.

The upregulation of components such as PSMC3 and PSMC1 signifies a robust enhancement of the ubiquitin-proteasome system [46], which is vital for the degradation of misfolded or damaged proteins. Coupled with proteins like ALYREF, TXN2, and THBS1, which are also upregulated, this activity emphasizes a cellular environment adapted to maintaining health amid oxidative stress and maintaining effective intercellular communication. In a 3D model that closely mimics physiological settings, this activation is critical for preventing protein accumulation that could lead to cellular dysfunction, particularly in neurons that face high metabolic demands.

Within the neuronal context, the upregulation of proteasome components, combined with proteins like ALYREF and TXN2, complements the roles they play in synaptic plasticity and cellular signaling. ALYREF facilitates mRNA export [47], ensuring that necessary transcripts are available for protein synthesis, which is essential during synaptic changes. TXN2 contributes to managing oxidative stress [48,49], protecting neurons from potential damage, and allowing for sustained neuronal activity. The increased expression of EIF3L may reflect an upsurge in translational efficiency, allowing for quick responses to cellular signals and demands, including the production of proteins necessary for cellular survival and adaptation. This efficiency is particularly important in neurons, where rapid local protein synthesis can be crucial for synaptic plasticity and the overall maintenance of neural networks [50,51].

Conversely, the downregulation of proteins such as FXR1, LMNB2, and RPS19 indicates a necessary transition as neurons mature and solidify their functional roles [52]. FXR1’s downregulation suggests reduced mRNA turnover needs [53], allowing for a more stable expression profile that is characteristic of differentiated cells, like neurons or neural cells. The downregulation of LMNB2 reveals potential changes in nuclear architecture [54] that reflect a move towards the functional stability necessary for specialized roles, while the decrease in ribosomal proteins like RPS19 points to a shift from protein synthesis towards maintaining established structures and functions. This balanced interplay of upregulated and downregulated proteins, including the proteasome complex, highlights the intricate regulatory networks that underpin neuronal stability, adaptability, and intercellular communication within the model, emphasizing their vital roles in maintaining the overall health and functionality of the nervous system.

The overall protein expression profile observed in the vitamin combination group indicates a balanced response, promoting protective and adaptive mechanisms while downregulating pathways inconsistent with mature neuronal function. This stands in contrast to the isolated vitamin B12 treatment, which may not provide the same comprehensive benefits as the combination, thus limiting the potential upregulation of protective proteins and failing to promote the necessary downregulation of others.

Both the combination treatment of vitamins, as well as the B12 treatment alone, led to significant upregulation of several key proteins, including MYL12B, MYEF2, EIF6, NUP98, IMMT, PSME3, TUBB2B, CSRP1, HEBP1, and PRDX6, compared to the control group. This suggests that both treatments activated biological processes such as protein synthesis, cellular signaling, mitochondrial function, and oxidative stress responses. It was observed that the combination vitamin treatment uniquely upregulated the proteins ALYREF, PSMC1, USP15, EIF3L, and THBS1 compared to both the B12 treatment alone and the control group (Figure 6). This specific upregulation highlights the unique synergistic effects of the combination treatment.

One of the objectives of these studies was to further investigate and establish in vitro models to study the pathophysiology of PN, the effect of treatment options, and their biochemical mode of action, so that animal testing could be avoided in the future. The 3D co-culture model was already successfully established by Smith and co-workers [16] and was reproduced and further developed in this study. Additionally, we demonstrated that this model is suitable to compare the effect of different treatment options, a combination of B1, B6, and B12 vs. each single B vitamin and with each other, and conduct the tests in a non-depleted state. The cell culture models (2D and 3D) are also considered applicable based on our results to demonstrate treatment effects on neural cell function, synapsing, cell maturation, formation of dendritic networks and regeneration. Further, these models provided the basis for a more detailed investigation of the biochemical mode of action of treatments using the proteomics analysis, which is a novel approach. To focus on oxidative stress as a main damaging factor in an in vitro model is reasonable, as the diverse upstream causes for PN (hyperglycemia, dyslipidemia, some chemotherapeutics, alcohol abuse) converge into one prominent metabolic disturbance: the production of ROS and, thus, increased oxidative stress. The most prevalent and most studied PN is DPN, in which oxidative stress, as pointed out earlier in the discussion, is one of the major factors driving neuronal damage. We decided, therefore, to test the potential regenerative effects of neurotropic B vitamins in an in vitro model based on hydrogen peroxide-induced neurodegeneration. A recently conducted prospective, placebo-controlled pilot study investigated the effects of a combination of vitamins, minerals, and other actives, which contained vitamins B1, B6, and B12. In the active group, patients with diabetic neuropathy reported a significant reduction in pain scores (PS from 20.9 to 13.9, *p* < 0.001; [55]). These promising clinical findings, in combination with the results of this in vitro study, may give occasion to further investigate the effectiveness of B vitamins for DPN. It is not possible to determine which substance exerted the reported effect in the pilot study by Didangelos and colleagues, as individual substances were not tested. However, this study contributes a biochemical mechanistic justification that neurotropic B vitamins play a major role in neuronal regeneration and, thus, could be used to treat PN symptoms.

This work also delivers a contribution in the context of the 3R principle. The goal of the 3R Principle is to avoid animal experiments altogether (Replacement) and to limit the number of animals (Reduction) and their suffering (Refinement) in tests to an absolute minimum. In biomedical sciences, alternatives to animal testing become more important as ethical considerations find more attention within and outside the scientific society [56,57]. The 3R Principle now provides a road map for laboratory animal protection policies of many countries. Besides in vitro cell culture models, organoids and simpler organisms, such as invertebrates, have been suggested to avoid classical animal testing with rodents or other mammals. Recently, computational models became more relevant in implementing the moral demands posted by the 3R Principle in that they provide the means to digitally scrutinize experimental setups. The number of animal experiments can be reduced with digital models which are based on real animal or in vitro models, like the one we studied in this work [56].

## 5. Conclusions

In conclusion, the findings provide a mechanistic rationale for the need and superiority of the vitamin B1, B6, and B12 combination over treatment with individual B vitamins. The hypothesis that the vitamins B1, B6, and B12 act in biochemical synergy in the nervous system by supplying energy, enabling signal transduction, and providing neuroprotection in the form of the myelin sheath is supported by numerous in vitro and animal studies [12]. As each vitamin serves a unique function by its individual mode of action, they cannot replace each other [13,58].

On the cellular level, key markers of neuronal functionality and viability, cell maturation, and enhancement of morphological features, like synapse formation and networking, were improved significantly compared to treatment with individual B vitamins. This holds up both in healthy and damaged cells. On the level of protein expression, the combination of vitamins B1, B6, and B12 uniquely elevated five proteins which are involved in gene regulation, protein homeostasis, and extracellular matrix remodeling. These processes are critical for neuronal function and repair and match the results we found on the cellular level in that they provide a molecular, biochemical explanation for the cellular processes. Before, an advantage of the combination of vitamins B1, B6, and B12 on neuronal parameters has only been demonstrated in a diabetic rat model of PN [43]. The three neurotropic B vitamins have demonstrated significant influence on various measures of cell viability and function in vitro. The observed effects of the combination of vitamins B1, B6, and B12 on damaged and healthy cells exceeded those of each individual B vitamin and proved the superiority of the combination for enabling regeneration. The combination of vitamins B1, B6, and B12 synergistically enhanced neural maturation and increased neuroplasticity and connectivity. The proteomic analysis provided completely novel insights and mechanistic details by elucidating the underlying molecular changes related to neuronal stability, adaptability, and intercellular communication within the model, emphasizing vital roles in maintaining the overall health and functionality of the nervous system. This provides a mechanistic rationale for the need and superiority of the combination of vitamins B1, B6, and B12 vs. individual B vitamins and a new insight into the molecular mode of action. The relatively simple but valid model setup bridges oxidative stress, the common metabolic cause of different PNs, with the neuro-regenerative effect of B vitamins and, thus, justifies the application of neurotropic B vitamins in clinical practice [10]. The study discussed here provides a rationale for including all three of these neurotropic B vitamins in a therapeutic product.

## Figures and Tables

**Figure 1 cells-14-00477-f001:**
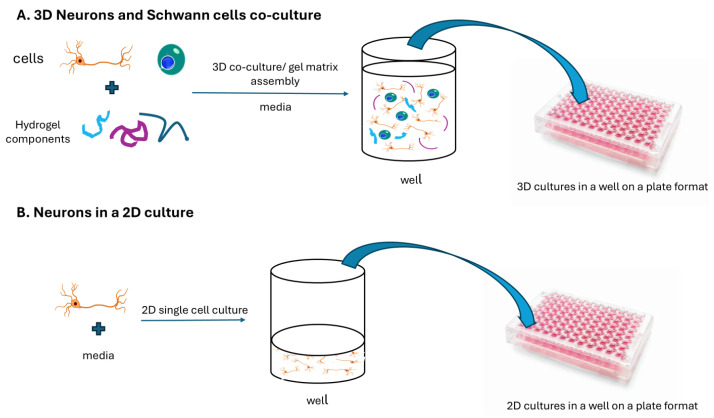
Two approaches for in vitro neuron cell cultures. (**A**) a three-dimensional assembly of neurons and Schwann cells embedded in a cellular matrix; (**B**) a single-cell two-dimensional culture.

**Figure 2 cells-14-00477-f002:**
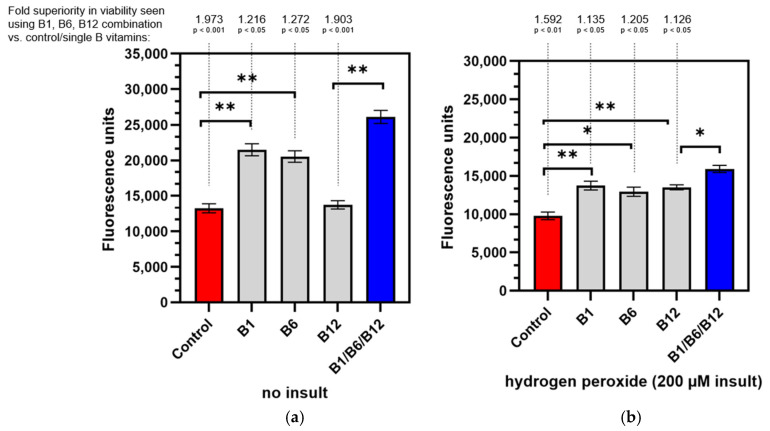
Viability of neuron and Schwann cell 3D co-cultures under non-insulted (**a**) and hydrogen peroxide-insulted (**b**) conditions. Non-insulted and insulted cells were treated with control medium, individual vitamins B1, B6, or B12, or the B1/B6/B12 combination followed by alamarBlue^®^ staining. Absolute fluorescence intensity is shown. *p*-values above the brackets: * *p* < 0.05 vs. control; ** *p* < 0.01 vs. control. *p*-values above the brackets indicate statistical significance vs. B1/B6/B12 combination. Control = no vitamin treatment, control medium with low-level B vitamins. Treatment: B1 = 40 mM vitamin B1; B6 = 20 µM vitamin B6; B12 = 0.4 µM vitamin B12; B1/B6/B12 combination = combination of vitamins B1 (40 µM), B6 (20 µM), and B12 (0.4 µM).

**Figure 3 cells-14-00477-f003:**
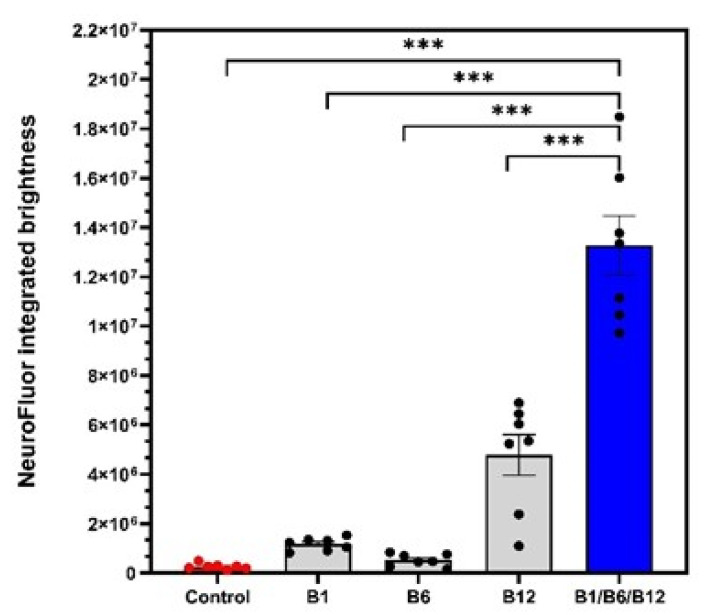
3D co-cultures of neurons and Schwann cells were treated with control medium, individual vitamins B1, B6, or B12, or the B1/B6/B12 combination followed by staining with NeuroFluor to test for neuronal maturation. Neuron maturation under different treatment conditions was measured as integrated brightness of green fluorescence. *p*-values above the brackets: *** *p* < 0.001 vs. control. *p*-values above the brackets indicate statistical significance vs. B1/B6/B12 combination. Control = no vitamin treatment, control medium with low-level B vitamins. Treatment: B1 = 40 mM vitamin B1; B6 = 20 µM vitamin B6; B12 = 0.4 µM vitamin B12; B1/B6/B12 combination = combination of vitamins B1 (40 µM), B6 (20 µM), and B12 (0.4 µM).

**Figure 4 cells-14-00477-f004:**
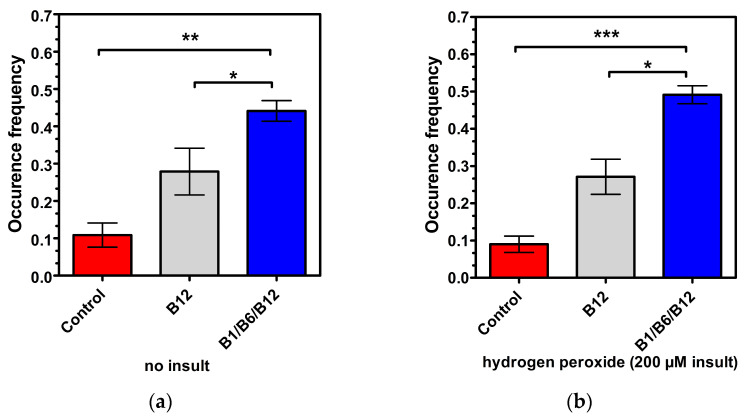
Synapsing and networking of the neuron cells (non-insulted (**a**) vs. insulted (**b**)) treated with control medium, vitamin B12 alone, or the B1/B6/B12 combination, followed by morphological analysis. *p*-values above the brackets: * *p* < 0.05 vs. control; ** *p* < 0.01 vs. control; *** *p* < 0.001 vs. control. *p*-values above the brackets indicate statistical significance vs. B1/B6/B12 combination. Control = no vitamin treatment, control medium with low-level B vitamins. Treatment: B12 = 0.4 µM vitamin B12; B1/B6/B12 combination = combination of vitamins B1 (40 µM), B6 (20 µM), and B12 (0.4 µM).

**Figure 5 cells-14-00477-f005:**
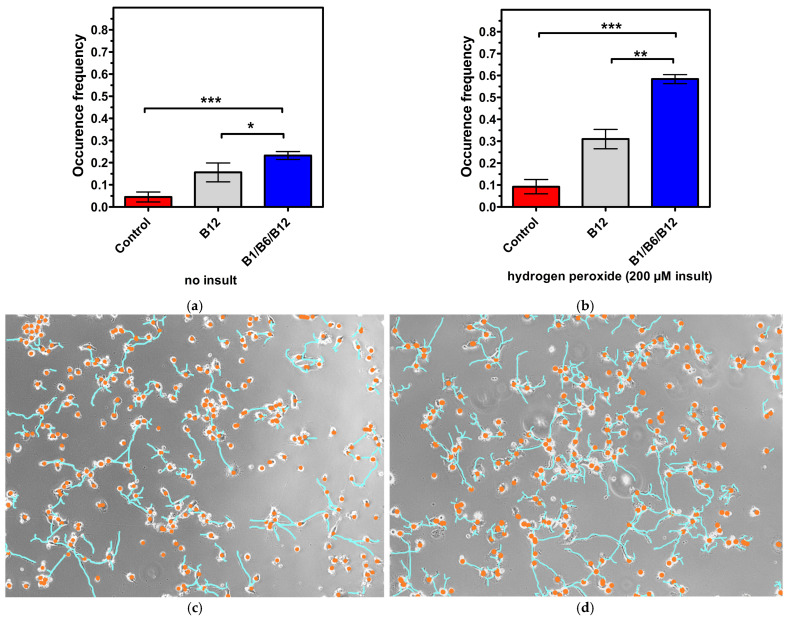
(**a**,**b**) Synapsing and networking of the neuron cells (non-insulted (**a**) vs. insulted (**b**)) treated with control medium, vitamin B12 alone, or the B1/B6/B12 combination, followed by morphological analysis. *p*-values above the brackets: * *p* < 0.05 vs. control; ** *p* < 0.01 vs. control; *** *p* < 0.001 vs. control. *p*-values above the brackets indicate statistical significance vs. B1/B6/B12 combination. (**c**,**d**) Live NG108 neuron images captured under phase contrast (non-insulted and treated with control medium (**c**) or insulted and treated with the B1/B6/B12 combination (**d**)). Neuron morphology in the micrographs was simplified, with dendrites represented as lines and cell bodies as circles. Control = no vitamin treatment, control medium with low-level B vitamins. Treatment: B12 = 0.4 µM vitamin B12; B1/B6/B12 combination = combination of vitamins B1 (40 µM), B6 (20 µM), and B12 (0.4 µM).

**Figure 6 cells-14-00477-f006:**
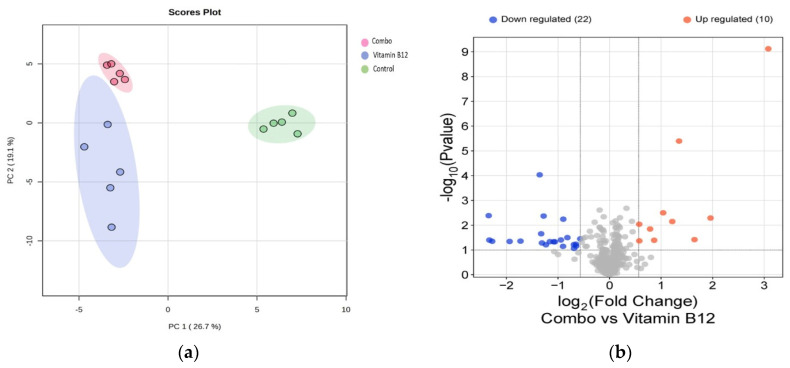
(**a**) PCA plot showing distinct clustering of the treatment groups, indicating significant differences in protein expression profiles; (**b**) Volcano plots highlighting upregulated proteins in the combination treatment versus B12.

**Table 1 cells-14-00477-t001:** Comparison of the effects of the vitamin B combination on nerve cell maturation compared to untreated control or individual vitamin treatment. All results are at *p* < 0.001 significance.

	Fold Superiority of B1, B6, B12 Combination Treatment on Nerve Cell Maturation
B1 alone	14.3-fold
B6 alone	50.9-fold
B12 alone	2.9-fold

**Table 2 cells-14-00477-t002:** Comparison of the effects of the vitamin B combination and vitamin B12 vs. untreated control, and direct comparison of the vitamin combination and vitamin B12 in supporting synapsing and networking.

	No Insult	H_2_O_2_ Insult
Superiority of the B1, B6, B12 combination vs. untreated control		
Synapsing	4-fold	5.44-fold
Networking	4.6-fold	6.44-fold
Superiority of B12 vs. untreated control		
Synapsing	2.55-fold	3-fold
Networking	3.2-fold	3.44-fold
Superiority of the B1, B6, B12 combination vs. B12 alone		
Synapsing	1.94-fold	2.22-fold
Networking	1.64-fold	2.23-fold

**Table 3 cells-14-00477-t003:** Biological processes activated by the B1/B6/B12 combination treatment vs. B12 alone—proteomics analysis.

FDR	Biological Processes/ Pathways	Related to Neuron	Proteins
8.25 × 10^−4^	Proteasome complex	Synaptic plasticity and maintaining cellular homeostasis	PSMC1 PSMA7 PSMD11 PSMD6 PSMD12
1.74 × 10^−3^	Synapse	Synaptic communication and neurotransmission	RPL38 RHOA PICALM CLTA SH3GL1 ARFGAP1 COPS4 YWHAZ PSMA7 EIF3A SPTBN1 HNRNPD IGF2BP1 FABP5 YWHAG DES RPL10A ITSN1
7.07 × 10^−8^	Axon guidance	Proper formation of neural circuits and connectivity	RHOA MYL6 PSMC1 PSMA7 MAPK3 PSMD11 SPTBN1 CLTA TLN1 MSN PSMD6 RPL38 PSMD12 SPTAN1 RPL10A ITSN1 RPS28 ARPC4
9.38 × 10^−8^	Cellular responses to stress	Maintenance of cellular homeostasis, protection from damage and promotion of cell survival	TPR ASNS XPO1 KHSRP PSMC1 PSMA7 ARFGAP1 MAPK3 PSMD11 LMNB1 TLN1 PSMD6 PRDX3 STAT3 HSPA4 RPL38 DCTN2 PSMD12 TXNRD1 RPL10A RPS28
1.38 × 10^−4^	Regulation of expression of SLITs and ROBOs	Establishment of proper neuronal connections and circuitry	PSMC1 PSMA7 PSMD11 PSMD6 RPL38 PSMD12 RPL10A RPS28
2.08 × 10^−3^	Signaling by NOTCH	Neuronal differentiation, maturation, and synaptic plasticity	SNW1 PSMC1 PSMA7 PSMD11 PSMD6 YWHAZ PSMD12
4.72 × 10^−4^	Signaling by WNT	Neuronal development and synaptic plasticity	RHOA XPO1 PSMC1 PSMA7 PSMD11 CLTA PSMD6 YWHAZ PSMD12

Top themes from the proteomics analysis of the 3D neuron/Schwann cell co-cultures treated with vitamin B12 alone or the B1/B6/B12 combination. Themes reflect key differences between the two treatments. FDR, false discovery rate; ROBO, roundabout (receptor of SLIT); WNT, wingless-related integration.

## Data Availability

The data presented in this study are available on request from the corresponding author (viel.c@pg.com).

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
