# Peer review of "The Combination of Neurotropic Vitamins B1, B6, and B12 Enhances Neural Cell Maturation and Connectivity Superior to Single B Vitamins"

_cells, 2025, doi:10.3390/cells14070477_

Round 1
Reviewer 1 Report
Comments and Suggestions for Authors
The study presented does follow a clear design and has sound results, but the presentation of these results as well as the drawn conclusion should be revised.The combined treatment appears to alleviate the results of oxidative stress insult in the cell culture model, but in my oppinion it is not clearly demonstrated why could this be translated to peripheral neuropathy.
It is not clearly substantiated, why oxidative stress is the dominant pathophysiology of PNP, and it is not clear why this in vitro setting is a valid model of the disease.
The description of the morphological analysis is quite vague and the presentation of the data is also somewhat lacking. Although the data of Figure 3 and 4 appear to be different, the legends are identical. What exact parameters were used in the morphometry? Could some results be visually demonstrated?It is also not indicated, how many cells or fields of views were measures. Although it is easy to guess from the colors, which bar belongs to which group, some labels are missing from the figures 1, 3 and 4. In line 475 do the authors mean the improvement after treatment is more pronounced in the damaged cell culture or was this measured at individuals damaged cells and compared to healthy looking cells from the same group? In line 481: where was the improvement of Schwann cell-neuron interactions demonstratred?
The volcano plot does not compare treatment v control and why were measurements from B1 and B6 vitamin treatments alone were omitted?
In line 490: in the cited paper shows no change in oxidative stress parameters in the animals which would question the validity of the purely oxidative stress based in vitro model.
Reviewer 2 Report
Comments and Suggestions for Authors
The authors reported the efficacy of combination treatment with vitamins B1, B6, and B12 for neural cell maturation and connectivity in vitro. While the manuscript is comprehensible, there are numerous concerns regarding the methodology, data presentation, and interpretation. In particular, morphological evidence should be added to support the quantitative data.
[Introduction]
- Since this is an original research article rather than a review, the use of subsections (1.1-1.4) should be avoided.
- The authors should specify the neuropathies for which vitamins therapy is applicable.
[Materials and Methods]
- A schematic representation of the 3D and 2D co-culture models would help readers better understand the methodology.
- Limited information is provided regarding the Schwann cell line (SCL4.1) and neuron line (NG108) (Page 3, Lines 112-113). In particular, the characteristic features of SCL4.1 cells (e.g. expression of glial cell markers, synthesis and secretion of neurotrophic factors, myelin formation in co-culture with neurons) should be described.
- Co-cultures were maintained in a medium containing 10% FBS (Page 3, Line 118), which may lead to excessive proliferation of the lined cells.
- The 2D co-culture model was used to evaluate synapsing and networking parameters (Page 3, Lines 134-135), but was it used not myelination?
[Results]
- The authors presented only statistical data, making it difficult to understand their experimental approach. Representative photomicrographs of cultured cells would help clarify the findings.
- The data in Table 1 and Fig.1 should be combined. It would be preferable to present viability as a fold increase rather than a percentage in Table 1. Additionally, the term [healthy cells] should be changed to [untreated cells].
- In Figs. 1, 3, and 4, the cytotoxicity of hydrogen peroxide should be more clearly indicated.
- Representative phase-contrast and/or immunocytochemical photomicrographs should be included to support the data on cell connectivity and maturation.
- It is unclear why both 2D and 3D co-culture models were used. Evaluating the efficacy of vitamin B combination therapy using individual neurons and Schwann cells might be more informative.
- Schwann cell maturation can be assessed by using marker proteins such as p75, L1, Krox 20, and SOX10.
- The proteomics data (Fig.5 and Table 4) may provide insights into the molecular mechanisms underlying the efficacy of the B1/B6/B12 combination treatment. However, additional experiments, such as Western blotting and pathway inhibition assays, would be desirable to further elucidate the involved signaling pathways.
[References]
The following article must be cited.
Didangelos T, et al., Efficacy and safety of the combination of palmitoylethanolamide, superoxide dismutase, alpha lipoic acid, vitamins B12, B1, B6, E, Mg, Zn and nicotinamide for 6 months in people with diabetic neuropathy. Nutrients 16:3045 (2024)
Round 2
Reviewer 2 Report
Comments and Suggestions for Authors
The manuscript has been revised in accordance with the reviewer’s comments; however, the following issues still need to be addressed.
[Materials and Methods]
- The authors should cite the following article, which describes the characteristic features of SCL4.1 cells, including myelin formation.
Haynes LW, et al., J Neurosci Methods 52:119-127 (1994).
- Regarding the potential FBS-induced proliferation of the cell lines, the authors stated that this variable is controlled and that the enhancement in viability with the triple vitamin combination surpasses both mono-vitamin treatments and untreated controls. It would be desirable for the authors to provide morphological data to support this statement.
[Results]
Some of the authors responses to the reviewer’s comments should be incorporated into the manuscript text.
